# Vitamin D Deficiency in Patients with Morbid Obesity before and after Metabolic Bariatric Surgery

**DOI:** 10.3390/nu14163319

**Published:** 2022-08-13

**Authors:** Mario Musella, Giovanna Berardi, Antonio Vitiello, Danit Dayan, Vincenzo Schiavone, Antonio Franzese, Adam Abu-Abeid

**Affiliations:** 1Advanced Biomedical Sciences Department, Naples Federico II University, AOU “Federico II”–Via S.Pansini 5, 80131 Naples, Italy; 2Division of General Surgery, Tel Aviv Sourasky Medical Center, 6 Weizman Street, 64230906, Affiliated to Sackler Faculty of Medicine, Tel Aviv University, 64230906 Tel Aviv, Israel

**Keywords:** metabolic bariatric surgery, severe obesity, vitamin D deficiency, sleeve gastrectomy, gastric bypass

## Abstract

Background: Metabolic bariatric surgery (MBS) is the most effective treatment for severe obesity. Vitamin D deficiency is a common complication encountered both during preoperative workup and follow-up. Aim: To estimate the prevalence of vitamin D deficiency in patients undergoing MBS. Methods: Prospectively maintained database of our university MBS center was searched to assess the rate of preoperative and postoperative vitamin D deficiency or insufficiency in patients undergoing MBS over a one-year period. Results: In total, 184 patients were included, 85 cases of Sleeve Gastrectomy (SG), 99 Gastric Bypass (GB; 91 One Anastomosis and 8 Roux-en-Y). Preoperative vitamin D deficiency and insufficiency were respectively found in 61% and 29% of patients, with no significant difference between SG and GB. After six months, 15% of patients had vitamin D deficiency, and 34% had vitamin D insufficiency. There was no significant difference in the rate of vitamin D deficiency or insufficiency and the percentage of total weight loss (%TWL) at 1, 3, and 6 postoperative months between SG and GB. Conclusions: Preoperative vitamin D deficiency or insufficiency is common in MBS candidates. Regular follow-up with correct supplementation is recommended when undergoing MBS. Early postoperative values of vitamin D were comparable between SG and OAGB.

## 1. Introduction

Morbid obesity continues to increase in western society, and it is associated with increased morbidity and mortality [1]. It is also related to nutritional disorders, with vitamin D deficiency or insufficiency being the most commonly reported [2,3,4].

To date, Metabolic Bariatric Surgery (MBS) serves as the most effective treatment for patients with morbid obesity as it is associated with long-term sustained weight loss, resolution of obesity-associated diseases, and decreased mortality [5,6]. The most commonly performed MBS reported in the 2018 International Federation for the Surgery of Obesity and Metabolic Disorders are Sleeve Gastrectomy (SG), Roux-en-Y Gastric Bypass (RYGB), and One Anastomosis Gastric Bypass (OAGB) [7]. Indications for MBS are Body Mass Index (BMI) of 40 kg/m^2^ or greater, BMI of 35 kg/m^2^ or greater with at least one obesity-associated co-morbidity, and BMI of 30 or greater for poorly controlled type 2 diabetes (T2D) [8]. There are several complications that may be associated with MBS, including those that may occur in the early (e.g., bleeding, leak) and late postoperative period (e.g., nutritional deficiency, marginal ulcer, chronic fistulas). The rate of total complications after MBS is reported to be 3.13%, and the rate of mortality is 0.16% [9].

Procedures with a malabsorptive component, such as OAGB and RYGB, have been reported to be associated with a higher risk of vitamin D deficiency when compared to solely restrictive procedures, such as SG [10].

MBS has been linked with postoperative nutritional deficiencies, mainly due to malabsorption and rapid weight loss, with vitamin D being the most prevalent [11,12,13].

Vitamin D deficiency has been shown to be associated with osteoporosis, muscle atrophy, recurrent falls, cancer, cardiovascular diseases, T2D, and increased respiratory infections [14].

In this study, we wanted to assess the prevalence of vitamin D deficiency or insufficiency in MBS candidates and analyze the effect of MBS on postoperative levels.

## 2. Methods

### 2.1. Patients and Data

A retrospective analysis of a prospectively maintained database of a single university metabolic bariatric center was performed to find all patients who had undergone surgery between January and December 2021. The following baseline, demographic, anthropometric, and clinical characteristics were retrieved from our database: Age, gender, body mass index (BMI), previous bariatric procedures, obesity-associated medical problems, preoperative and postoperative serum levels of vitamin D at 1, 3, and 6 months.

Weight loss was calculated as the percentage of total weight loss (%TWL):$$\frac{\mathrm{Initial}\;\mathrm{weight}\;-\;\mathrm{followup}\;\mathrm{weight}}{followup\;weight}\;\times \;100$$

### 2.2. Definition of Vitamin D Deficiency

Vitamin D deficits were defined in accordance with the most recent published guidelines of the ASMBS [15]:Vitamin D insufficiency—20–30 ng/mLVitamin D deficiency—<20 ng/mL

### 2.3. Preoperative Evaluation

All patients underwent preoperative multidisciplinary evaluation and met the American Society of Metabolic and Bariatric Surgery (ASMBS) guidelines:Metabolic bariatric surgeon—Assessment of BMI, obesity-associated medical problems, indication for MBS, type of MBS, referral to blood work (complete blood count, full chemistry panel including micronutrient levels, coagulation studies, thyroid function, cortisol levels), upper gastrointestinal (GI) endoscopy, and/or upper GI series.Endocrinologist—All patients were assessed by an endocrinologist for obesity-associated medical problems evaluation and their optimization preoperatively. Patients are also assessed for secondary causes of obesity.Dietitian—All patients were assessed by a dietitian preoperatively and were recommended a protein-rich, low-carbohydrate diet preoperatively.Psychologist/Psychiatrist—All patients underwent psychologic evaluation to ensure their understanding of lifestyle changes that are associated with MBS, the importance of compliance to follow-up, and medical supplementation.Other disciplines—Patients were referred to other disciplines for further evaluation and optimization if other morbidities (such as cardiovascular, respiratory, neurologic, etc.) exist.

Following the aforementioned evaluation, all patients were referred to a multidisciplinary committee that approves the indication for MBS.

### 2.4. Surgical Technique

During the study period, three surgical procedures were performed: Laparoscopic Sleeve Gastrectomy (SG), Laparoscopic One Anastomosis Gastric Bypass (OAGB), and Laparoscopic Roux-en-Y Gastric Bypass (RYGB).

SG—The omentum was dissected off the greater curvature of the stomach starting 4 cm proximal to the pylorus up to the Angle of His. A 36-F bougie was inserted along the lesser curvature for calibration, and the stomach was vertically transected with a linear stapler to create the gastric sleeve [16,17].OAGB—A 15–20 cm long gastric pouch was created with a linear stapler along a 36-Fr bougie for calibration. The bowel length was measured starting from the ligament of Treitz, and the anastomosis was performed 160–220 cm distal to the ligament of Treitz. The length of the biliopancreatic limb was fashioned according to the BMI, surgeons‘ preference, and indication of OAGB (primary or secondary for failed procedure or late complications). Prior to performing the anastomosis, it was verified that at least 300 cm bowel was present distally [18].RYGB—A 4–6 cm long gastric pouch was created with a linear stapler along a 36-Fr bougie for calibration. The bowel length was measured starting from the ligament of Treitz. The bowel was transected at 100 cm distal to the ligament of Treitz, defining the length of the biliopancreatic limb, the gastro-jejunal anastomosis was performed and the jejuno-jejunal anastomosis was performed 100 cm distal to it.

### 2.5. Postoperative Care

All patients received high-dose proton pump inhibitors (PPI) (40 mg twice per day) for 3 months and continued venous thromboembolism (VTE) prophylaxis with subcutaneous injection of enoxaparin 40 mg for three weeks. Multivitamin supplement, vitamin D drops (2000 IU/day), and Calcium-citrate (600mg/day) are prescribed uniformly to all patients. Following discharge, all patients underwent routine follow-up at the ambulatory clinic postoperatively at 2 weeks, 1 month, 3 months, 6 months, 1 year, and then annually.

### 2.6. Statistical Analysis

Statistical analysis was performed using IBM SPSS version 27. For normally distributed parameters, Student’s t-test and chi-square test for categorical data were performed. Continuous data are expressed as mean values ± standard deviation. All *p* values were derived from 2-tailed tests and they were considered significant when *p* value was <0.05.

## 3. Results

A total of 214 patients underwent MBS during the study period. Data were not available for 30 patients due to loss to follow-up, thus, 184 patients were included in our analysis: SG was performed in 85 patients, One Anastomosis Gastric Bypass (OAGB) in 91 patients, and Roux-en-Y Gastric Bypass (RYGB) in 8 patients. Since the number of patients in the RYGB group was small, they were added to the OAGB group and were named the Gastric Bypass (GB) group as OAGB and RYGB are considered combined restrictive and malabsorptive procedures, while SG is considered purely restrictive [19]. The baseline characteristics of patients are shown in Table 1. There were no statistically significant differences between SG and Gastric bypass (GB) in terms of age, gender, T2D, obstructive sleep apnea, dyslipidemia, and osteoarthritis. Patients undergoing GB had a statistically significant higher preoperative BMI, hypertension (HTN), gastroesophageal reflux disease (GERD), and a previous bariatric procedure when compared to SG (47.11 ± 7.39 vs. 41.04 ± 4.61; *p* < 0.001, 39% vs. 25%; *p* = 0.03, 20% vs. 1%; *p* < 0.001, and 15% vs. 2%, respectively). In the entire cohort, there were no patients suffering from chronic kidney disease or advanced liver disease.

The serum vitamin D levels in different periods are depicted in Table 2. The mean levels of vitamin D of the whole study group at baseline (for patients with at least 1 month follow-up of vitamin D level), 1 month, 3 months, and 6 months was, 18.07 ± 8.80, 20.76 ± 8.37, 25.32 ± 10.20, and 29.47 ± 11.41, respectively, with no statistically significant difference between SG to GB. When comparing baseline preoperative vitamin D levels for patients with at least 1 month follow up levels we found postoperative levels to be significantly higher at 1- (*p* = 0.02), 3- (*p* < 0.001, and 6-months (*p* < 0.001) for the whole study group.

Vitamin D deficits in different periods are depicted in Table 3. Preoperative vitamin D deficiency and insufficiency were respectively found in 61% and 29% of patients, with no significant difference between SG and GB. For patients with at least one month follow-up of vitamin D levels, preoperative vitamin D defi-ciency and insufficiency were respectively found in 67% and 30% of patients with no significant difference between SG and GB. For statistical analysis we compared postoperative vitamin D levels to preoperative vitamin D levels of patients with at least 1 month follow-up of vitamin D level. After 1, 3 and 6 months, 50%, 30% and 15%, had vitamin D deficiency, and 34%, 42%, and 34% had vitamin D insufficiency, respectively. Vitamin D deficiency was not significantly lower at 1 month postoperatively (*p* = 0.34) for SG, it was significantly lower at 3- (*p* = 0.003) and 6- months (*p* < 0.001) postoperatively for SG when compared to the preoperative status. For GB, vitamin D deficiency was significantly lower at 1- (*p* = 0.003), 3- (*p* < 0.001) and 6-months (*p* < 0.001) postoperatively. Vitamin D insufficiency was not significantly different at the all-follow-up periods for SG and GB when compared to the preoperative status (*p* > 0.05). There was no statistically significant difference in vitamin D deficiency or insufficiency and %TWL at 1, 3, and 6 months postoperatively between SG and GB.

## 4. Discussion

Vitamin D deficiency is very commonly diagnosed during the preoperative clinical assessment for MBS in patients with severe obesity [3,4,20]. The inverse relation of BMI and vitamin D levels have been reported, and it is theorized that this is due to altered dietary habits, avoidance of sun exposure, reduced bioavailability of vitamin D due to its sequestration on adipose tissue, and decreased hepatic 25-hydroxylase in obese patients with non-alcoholic fatty liver disease [20].

The rate of vitamin D deficiency in the preoperative period varies in different studies. Pellegrini et al. [4] reported at least one micronutrient deficiency present in 85.5% of MBS candidates, and vitamin D deficiency occurred in 74.5% of patients, being the most common micronutrient deficient. In a prospective cross-sectional design performed in Singapore [21], a total of 75.7% of patients had vitamin D deficiency and 20.7% had vitamin D insufficiency. In this study, they interestingly note that despite being a country close to the equator with marked sun exposure throughout the year, the rate of deficiency was still high. In addition, it was interestingly found that a higher degree of deficiency occurred in women and in patients with higher BMI, similar to other Asian studies. Similarly, Entrenas et al. [22] reported vitamin D deficiency to occur in 72% and insufficiency to occur in 28%. In a French study, Ducloux et al. [23] showed vitamin D deficiency in about 96% of patients, which is much higher than the other studies cited, they correlate that the difference could be explained by variable regional ultra-violet exposures or regional dietary differences in diet. Daniel et al. showed patients with severe obesity who were black or Asians are more prone to vitamin D deficiency [24]. In our study, the prevalence of preoperative vitamin D deficiency was 61%, and insufficiency was 29%, which is slightly lower than the reported literature. Severe obesity strongly correlates with vitamin D deficiency, but there are several other multifactorial etiologies that may affect these levels, such as region, ethnicity, and sun exposure.

MBS has been linked with postoperative malnutrition because of reduced intake of food, reduced intake of drugs, malabsorption of nutrients after MBS with a malabsorptive component, changes in eating patterns, and non-compliance with dietary and supplement recommendations [14]. Despite that, several reports have shown that MBS causes a gradual improvement in vitamin D levels. Arias et al. showed that the rates of vitamin D deficiency decreased over time in RYGB patients when compared with the preoperative assessment: 50% vs. 74% at 1 year; *p* < 0.001, 45% vs. 74% at 2 years; *p* < 0.002 and 41% vs. 74% at 3 years; *p* < 0.04 [25]. Krzizek et al. reported that vitamin D deficiency decreased from 93.9% preoperatively to 70.8%, 67.0%, and 57.0% at 1, 2, and 3 years, respectively. In addition, they showed that patients who underwent RYGB were more likely to suffer from vitamin D deficiency throughout the three-year follow-up when compared with SG [26]. Araújo et al. reported that the prevalence of vitamin D deficiency was 25.0%, and insufficiency was 51.9% at a mean of 8.7-year follow-up after RYGB [27]. Their study results also suggested that vitamin D deficiency was associated with discontinued follow-up service for MBS patients in the late postoperative period, higher BMI, lower total usual vitamin D intake (from food and supplementation), and lower UVB radiation levels. In our study, the rate of vitamin D deficiency reduced from 62% to 16% and vitamin D insufficiency increased from 25% to 33% in a 6-month period, which probably increased due to patients down-grading from deficiency to insufficiency. We also did not show any significant difference between SG and GB.

The main causes suggested to determine vitamin D deficiency following MBS are preoperative deficiency, inadequate supplementation, lack of follow-up, bile salt deficiency that may occur in malabsorptive procedures, bacterial overgrowth, and delayed blend of vitamin D ingested with bile acids and pancreatic enzymes [10]. The compliance with nutritional supplementation following MBS is crucial. In a recently published study by Steenackers et al., a questionnaire was developed and completed by 402 subjects who underwent MBS, 17.2% did not consume any nutritional supplementation. Increasing age and medicine were positive predictors of compliance, and negative predictors were forgetfulness, high price, and side effects experienced [12]. There are several guidelines regarding the treatment of vitamin D deficiency in patients with severe obesity pre- and post-MBS [28]. We recommend strictly following these guidelines to prevent the negative outcomes that may be associated with vitamin D deficiency.

## 5. Conclusions

Preoperative vitamin D deficiency or insufficiency frequently occurred in our group of MBS candidates. Regular follow-up with correct supplementation should be recommended for all patients undergoing MBS. Early postoperative values of vitamin D were comparable between SG and OAGB.

## Figures and Tables

**Table 1 nutrients-14-03319-t001:** Baseline characteristics of patients undergoing metabolic bariatric surgery.

Characteristic	Sleeve Gastrectomy	Gastric Bypass	*p* Value
Age, mean ± SD, years	37.77 ± 12.21	39.78 ± 10.78	0.23
Women, n (%)	66 (78%)	64 (65%)	0.053
Preoperative BMI kg/m^2^	41.04 ± 4.61	47.11 ± 7.39	<0.001
T2D, n (%)	9 (11%)	17 (17%)	0.20
HTN, n (%)	21 (25%)	39 (39%)	0.03
OSA, n (%)	6 (7%)	9 (9%)	0.61
Dyslipidemia, n (%)	8 (9%)	8 (8%)	0.75
GERD, n (%)	1 (1%)	20 (20%)	<0.001
Osteoarthritis, n (%)	8 (9%)	5 (5%)	0.25
Previous bariatric procedure, n (%)	2 (2%)	15 (15%)	<0.001

SD: Standard deviation; BMI: Body mass index; T2D: Type 2 diabetes; HTN: Hypertension; OSA: Obstructive sleep apnea; GERD: Gastroesophageal reflux disease.

**Table 2 nutrients-14-03319-t002:** Serum vitamin D levels in patients undergoing metabolic bariatric surgery.

Variable	Sleeve Gastrectomy	Gastric Bypass	*p* Value
Preoperative levels, mean ± SD (ng/ml) *	18.00 ± 8.96	18.15 ± 8.71	0.92
Postoperative levels at 1 month, mean ± SD (ng/mL)	20.70 ± 8.15	20.81 ± 8.70	0.94
Postoperative levels at 3 months, mean ± SD (ng/mL)	25.19 ± 11.33	25.47 ± 8.91	0.89
Postoperative levels at 6 months, mean ± SD (ng/mL)	30.18 ± 12.82	28.69 ± 10.25	0.53
Preoperative levels versus postoperative levels at 1 month **			0.02
Preoperative levels versus postoperative levels at 3 months **			<0.001
Preoperative levels versus postoperative levels at 6 months **			<0.001

SD: Standard deviation. * Patients with at least one month follow-up of vitamin D level. ** The analysis was done for the whole study group (SG+GB).

**Table 3 nutrients-14-03319-t003:** Vitamin D deficits in patients undergoing metabolic bariatric surgery.

Timing	Outcomes	Sleeve Gastrectomy	Gastric Bypass	*p* Value
Preoperative *	Vitamin D deficiency, n (%)	33/56 (59%)	39/51 (76%)	0.054
	Vitamin D insufficiency, n (%)	16/56 (29%)	16/51 (31%)	0.75
Postoperative one month	Vitamin D deficiency, n (%)	28/56 (50%)	25/51 (49%)	0.92
	Vitamin D insufficiency, n (%)	20/56 (36%)	16/51 (31%)	0.63
	%TWL, mean ± SD	11.07 ± 5.23	11.12 ± 4.72	0.94
Postoperative three months	Vitamin D deficiency, n (%)	17/54 (31%)	12/42 (29%)	0.45
	Vitamin D insufficiency, n (%)	20/54 (37%)	20/42 (48%)	0.71
	%TWL, mean ± SD	20.85 ± 11.25	21.03 ± 5.95	0.90
Postoperative 6 months	Vitamin D deficiency, n (%)	8/49 (16%)	6/45 (13%)	0.68
	Vitamin D insufficiency, n (%)	16/49 (33%)	16/45 (36%)	0.76
	%TWL, mean ± SD	30.24 ± 7.94	31.06 ± 9.01	0.57

TWL: Total weight loss; SD: Standard deviation. * For patients with at least one-month follow-up of vitamin D level.

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
