# Peer review of "Vitamin D Deficiency in Patients with Morbid Obesity before and after Metabolic Bariatric Surgery"

_nutrients, 2022, doi:10.3390/nu14163319_

Round 1
Reviewer 1 Report
This manuscript investigated the prevalence of vitamin D deficiency or insufficiency in MBS patients. This study was well designed and conducted. The data presented in the study is interesting to readers, but there are still some concerns:
1. The introduction was not sufficient. It would be reasonable if the information about three surgical procedures in the method is moved to the introduction section. The current research data about the vitamin D deficiency or insufficiency related to three surgical procedures should be provided in the introduction too.
2. The reason to the combination of two groups in line 119-120 is not sensible enough. Scientific fact-based reasons are necessary for this combination.
3. The results lack of statistical analyses for the difference of vitamin D deficiency or insufficiency between the pre-and post-operation although it is discussed in the discussion section.
4. The list of study results from the previous studies in the discussion section is redundant. It would be more interesting if the possible reasons for the disparities of the findings are discussed detailly.
5. There are numerous grammar mistakes in the manuscript.
Author Response
Reviewer #1: This manuscript investigated the prevalence of vitamin D deficiency or insufficiency in MBS patients. This study was well designed and conducted. The data presented in the study is interesting to readers, but there are still some concerns:
- The introduction was not sufficient. It would be reasonable if the information about three surgical procedures in the method is moved to the introduction section. The current research data about the vitamin D deficiency or insufficiency related to three surgical procedures should be provided in the introduction too.
- The reason to the combination of two groups in line 119-120 is not sensible enough. Scientific fact-based reasons are necessary for this combination.
- The results lack of statistical analyses for the difference of vitamin D deficiency or insufficiency between the pre-and post-operation although it is discussed in the discussion section.
- The list of study results from the previous studies in the discussion section is redundant. It would be more interesting if the possible reasons for the disparities of the findings are discussed detailly.
- There are numerous grammar mistakes in the manuscript.
Response:
Comment 1 - In accordance with the reviewer’s comment we added information about the three common surgical procedures to the introduction, the indications, and the complications. (Lines 34-45)
Comment 2 – As the RYGB group was small (n=8), we decided to add them to the OAGB group and name it the Gastric Bypass (GB) group. OAGB and RYGB are considered combined malabsorptive and restrictive procedures while SG is considered purely restrictive. We clarified this in the results section (Lines 124-127).
Comment 3- We thank the reviewer for this valuable comment, and we added the statistical analysis for vitamin D deficiency and insufficiency comparing the preoperative status to the postoperative status in different time periods in the results section (Lines 141-144).
Comment 4- We thank the reviewer for the comment. The discussion section in our manuscript reports the experience of other studies of vitamin D deficiency in the preoperative setting to the postoperative settings and we also included our results which are roughly similar to other studies published. We also noted that other studies may have different results due to various definitions for vitamin D deficits (deficiency vs insufficiency) and not all studies were sticking with the updated guidelines.
Comment 5- The grammatical errors were revised thoroughly in the manuscript.
Reviewer 2 Report
Mario Musella et al. performed experiments that allow checking vitamin D levels in patients with morbid obesity before and after metabolic bariatric surgery. The authors analyzed 184 patients treated with sleeve gastrectomy, gastric bypass.
Before bariatric surgery, vitamin D deficiency was found in 61% of patients treated with sleeve gastrectomy and 29% of patients treated with gastric bypass, but these data were not confirmed statistically. Sixt months follow-up reveals lower vitamin D deficiency in tested morbidly obese patients.
The paper has scientific value, and after addressing a few major and minor comments.
Major comments
· Line 15
“….Vitamin D deficiency or insufficiency 15 in patients undergoing MBS over a 1-year period.”
Why the authors did not show the data after one year, only after six months. Maybe the vitamin D concentration after one year is correct….
· Please provide kidney and liver diseases in patients’ description in Tab.1. These are the crucial organs for vitamin. D metabolism.
· Line 22-24
“Conclusions: Regular follow-up with correct supplementation significantly reduces the rates of Vitamin D insufficiency and deficiency in MBS candidates. Early postoperative values of vitamin D were comparable between SG and OAGB.”
I think authors should refer only to their data and not speculate about vitamin D supplementation, which is fat-soluble and its absorption depends on many factors, especially in postbariatric patients with dyslipidemia as shown in Tab.1
· Line 68-71
“Dietitian – All patients are assessed by a dietitian preoperatively and are recommended on a protein-rich, low-carbohydrate diet preoperatively. Patients are assessed for micronutrient deficiencies preoperatively and are administered vitamin supplementation if these exist.”
Line 106-107
“…..multivitamin supplement, vitamin drops (2000 IU/day), and Calcium-citrate (600mg/day).”
Does it mean that patients with low vitamin D before operation (61% of patients treated with SG and 29% with GB) were supplemented with vitamin D? Be more precise about the time of supplementation and the dose. It is challenging to assess vitamin D when patients are supplemented.
· Please provide in Tab.2 more precise statistical analysis concerning vitamin D. The readers have to know the difference between patients before bariatric surgery (before SG vs. before GB) and post bariatric surgery during the time 1, 3, and 6 months. Is it statistically significant 16,8 ng/ml vs. 30.1 (SG before vs. 6 months after) or 17.3ng/ml vs. 28.6 (GB before vs.6 months after)?
· Provide more biochemical and metabolic parameters connected with vitamin D: lipids (patients have statistically significant dyslipidemia), calcium, phosphorus, and magnesium. Maybe detected vitamin D level is enough to keep calcium and phosphorus homeostasis???
Minor comments
· I think the introduction and discussion should be enriched with more details concerning vitamin D. Write about who is qualified for the MBS (BMI, etc.) and the risk of bariatric surgery. What deficiencies may appear after surgery, apart from vitamin D? Why do the authors get interested in vitamin D and not hypoglycemia, which occurs in almost 50% of postbariatric patients?
· When introducing abbreviations in the abstract, use them constantly instead of full names.
· Line 128
“The serum vitamin D levels in different periods are depicted in Table 2 and Figure 1.”
The data should not duplicate either table or figure
· Explain the abbreviation PPI, VTE
Author Response
Reviewer #2: Mario Musella et al. performed experiments that allow checking vitamin D levels in patients with morbid obesity before and after metabolic bariatric surgery. The authors analyzed 184 patients treated with sleeve gastrectomy, gastric bypass.
Before bariatric surgery, vitamin D deficiency was found in 61% of patients treated with sleeve gastrectomy and 29% of patients treated with gastric bypass, but these data were not confirmed statistically. Sixt months follow-up reveals lower vitamin D deficiency in tested morbidly obese patients.
The paper has scientific value, and after addressing a few major and minor comments.
Major comments
- 1. Line 15
“….Vitamin D deficiency or insufficiency 15 in patients undergoing MBS over a 1-year period.”
Why the authors did not show the data after one year, only after six months. Maybe the vitamin D concentration after one year is correct….
- 2. Please provide kidney and liver diseases in patients’ description in Tab.1. These are the crucial organs for vitamin. D metabolism.
- 3. Line 22-24
“Conclusions: Regular follow-up with correct supplementation significantly reduces the rates of Vitamin D insufficiency and deficiency in MBS candidates. Early postoperative values of vitamin D were comparable between SG and OAGB.”
I think authors should refer only to their data and not speculate about vitamin D supplementation, which is fat-soluble and its absorption depends on many factors, especially in postbariatric patients with dyslipidemia as shown in Tab.1
- 4. Line 68-71
“Dietitian – All patients are assessed by a dietitian preoperatively and are recommended on a protein-rich, low-carbohydrate diet preoperatively. Patients are assessed for micronutrient deficiencies preoperatively and are administered vitamin supplementation if these exist.”
Line 106-107
“…..multivitamin supplement, vitamin drops (2000 IU/day), and Calcium-citrate (600mg/day).”
Does it mean that patients with low vitamin D before operation (61% of patients treated with SG and 29% with GB) were supplemented with vitamin D? Be more precise about the time of supplementation and the dose. It is challenging to assess vitamin D when patients are supplemented.
- 5. Please provide in Tab.2 more precise statistical analysis concerning vitamin D. The readers have to know the difference between patients before bariatric surgery (before SG vs. before GB) and post bariatric surgery during the time 1, 3, and 6 months. Is it statistically significant 16,8 ng/ml vs. 30.1 (SG before vs. 6 months after) or 17.3ng/ml vs. 28.6 (GB before vs.6 months after)?
- Provide more biochemical and metabolic parameters connected with vitamin D: lipids (patients have statistically significant dyslipidemia), calcium, phosphorus, and magnesium. Maybe detected vitamin D level is enough to keep calcium and phosphorus homeostasis???
Minor comments
- I think the introduction and discussion should be enriched with more details concerning vitamin D. Write about who is qualified for the MBS (BMI, etc.) and the risk of bariatric surgery. What deficiencies may appear after surgery, apart from vitamin D? Why do the authors get interested in vitamin D and not hypoglycemia, which occurs in almost 50% of postbariatric patients?
- When introducing abbreviations in the abstract, use them constantly instead of full names.
- Line 128
“The serum vitamin D levels in different periods are depicted in Table 2 and Figure 1.”
The data should not duplicate either table or figure
- Explain the abbreviation PPI, VTE
Response: Major comments
Comment 1: In this study we analyzed patients undergoing MBS in 2021, when preparing this manuscript, most of the patients did not have a one-year follow-up and therefore we included data regarding 6 months follow-up
Comment 2: We thank the reviewer for this valuable comment, in our cohort there were no patients with chronic kidney disease and no patients with advanced liver disease. We added a clarification to this on our results section (Lines 132-133).
Comment 3: We agree with the reviewer’s comment, and we removed the statement regarding vitamin supplementation from our conclusions. We changed it to a more guarded comment stating “Regular follow-up with correct supplementation should be recommended to all patients undergoing MBS”. We also changed this in the abstract conclusion.
Comment 4: We thank the reviewer for this valuable comment and in accordance with this we have removed this confusing statement regarding preoperative treatment. As noted in the methods (lines 112-113) patients are prescribed vitamin D supplements in the postoperative setting.
Comment 5: We have made the statistical analysis in accordance with the reviewer’s comment. We added the analysis to our results section – lines 141-144.
“Vitamin D deficiency was significantly lower at 3 and 6 months postoperatively (p<0.0001) for SG and GB when compared to the preoperative status. Vitamin D insufficiency was not significantly different at the all-follow-up periods for SG and GB when compared to the preoperative status(p>0.05).”
Comment 6 – We agree with the reviewer’s comment vitamin levels may be affected by several parameters, however, in this brief report, we could not include these parameters.
Response: Minor comments
Comment 1- We enriched the introduction with the with indications for MBS as well as complications (lines 34-45). In this brief report, we chose to focus on vitamin D deficits solely. We used the abbreviations in the abstract after introduced in accordance with the reviewer’s comment (Line 21)
Comment 2 – Following the reviewer’s comment we have deleted the figure
Comment 3 – We have corrected the text and named the abbreviations VTE as Venous thromboembolism and PPI and proton pump inhibitors. Lines 110-111
Round 2
Reviewer 2 Report
Dear Authors, thank you for improving your manuscript.
Author Response
Dear editor,
Thank you for considering our manuscript for publication
We have replied to all comments point by point.
1-There are no significative statistical difference between Vit D levels, vitamin D deficiency and insufficiency, previous or during follow-up, between SG and GB, and so Table 2 and Table 3 have not to include differents rows with the different tecniques.
Response- In Table 1 we addressed the mean levels of vitamin D during the preoperative status and during the follow-up period. We think that it is important to show how levels of vitamin D increase during the follow-up period and that there is no difference between SG and GB as GB is considered by some to cause more vitamin D deficiency. We also added the p value comparing the preoperative vitamin D levels to 1-, 3-, and 6- months follow-up and showed that it significantly increased in all periods. In table 3 we specifically addressed the number of patients with deficiency and insufficiency according to the ASMBS guidelines. We think it is important to emphasize these definitions and show the results accordingly. In addition, we added the percentage of total weight loss to this table.
We think that presenting all the data in these ways are interesting to the readers, the tables contain different data, Table 1 shows clearly the levels of vitamin D in all periods, comparing SG to GB, comparing preoperative to postoperative levels, and Table 2 shows the number of patients with vitamin D deficits in all periods.
2- The preoperative Vit D levels were measured in 184 patients, at 1, 3 and 6 months we have Vit D levels in 107, 96 and 94 patients respectivaly. 2.1-Table 2: We can not use the T of Student test to compare, only the T of Student Test for related samples and include in the analysis only patients with the whole follow-up . Please add total sample values, but only with patients with at least 1 month values, and compare with the T of Student Test for related samples 2.2- The same consideration with the %TBWL 2.3- Table 3: only include preoperative % of patients with at least 1 month of follow-up
Response- We thank the editor for this valuable comment
2.1 – In Table 2 we changed the preoperative values to patient with at least one month follow-up.
2.2 For %TWL, the values given are concerning patients with at least one month follow-up.
2.3- We changed the levels of vitamin D deficiency and insufficiency in the table and included patients with at least one month follow-up to table 3.
3- In the conclusión you say: Regular follow-up with correct supplementation significantly reduces the rates of these disorders, but In the postoperative care you have not clarify if Vit D suplementation was different in patients with previously vitamin D deficiency and insufficiency that in normal Vit D level patients.
Response –In our revised manuscript we changed the statement “Regular follow-up with correct supplementation significantly reduces the rates of these disorders” to a more guarded statement – “Regular follow-up with correct supplementation should be recommended to all patients undergoing MBS”. In addition, in the - methods section, postoperative care, we clarified that all patients receive the same vitamin supplementations postoperatively (line 115-116).
4- Table 2 Reviewe 2r : Please provide in Tab.2 more precise statistical analysis concerning vitamin D. The readers have to know the difference between patients before bariatric surgery (before SG vs. before GB) and post bariatric surgery during the time 1, 3, and 6 months. Is it statistically significant 16,8 ng/ml vs. 30.1 (SG before vs. 6 months after) or 17.3ng/ml vs. 28.6 (GB before vs.6 months after)? You have not included p value between initial and follow-up Vit D values, only between both procedures
Response- We thank the reviewer for the comment and for the editor for lighting this up. We agree that showing the difference of vitamin D levels before and after bariatric surgery is important., we made the analysis and added it to the results section (lines 138-141, and 144-156) and added it to table 2.